# OpenReview forum: "Persuade Me if You Can: A Framework for Evaluating Persuasion Effectiveness and Susceptibility Among Large Language Models"
_ICLR.cc/2026/Conference — Submitted to ICLR 2026_

### Official Review · Reviewer_ADkH · 2025-11-01

**Soundness:** 3
**Presentation:** 3
**Contribution:** 3
**Rating:** 6
**Confidence:** 3

**Summary:**

The paper introduces an automated framework for measuring both persuasiveness and susceptibility to persuasion in large language models (LLMs). The framework runs multi-turn simulated dialogues between a PERSUADER and a PERSUADEE, quantifying (i) how effectively models change others’ opinions and (ii) how resistant they are to persuasion. The work targets scalability by replacing costly human annotation with automated LLM evaluations, while still claiming human-aligned validity.

Across tested models, Llama-3.3-70B and GPT-4o achieve similar persuasive strength, both outperforming Claude 3 Haiku by almst 30 %.

**Strengths:**

Good problem statement: Persuasion and susceptibility are central to LLM safety and alignment; a unified framework for both is a meaningful step forward.

The dual-role simulation (persuader/persuadee) is clearly structured and allows exploration of both offensive and defensive persuasion capabilities. The evaluation spans multiple foundation models and includes both subjective and misinformation scenarios, providing useful descriptive insights.

Even limited, the correlation study between automated scores and human judgments increases credibility relative to purely synthetic setups

**Weaknesses:**

Persuasiveness and susceptibility are measured entirely using LLM audiences. Prior studies Singh et al. (2024), Anthropic (2024), Hackenburg et al. (2024) demonstrate that LLMs are poor proxies for human persuasion, and that effectiveness scales logarithmically with model size.
The study does not evaluate on real-world persuasion datasets (e.g., ChangeMyView, PersuasionArena), leaving open whether the simulated results translate to authentic human discourse.
It is a promising framework conceptually, offering a scalable lens on persuasion dynamics. However, its empirical claims rest heavily on simulated LLM interactions,

**Questions:**

NA

---

> ### Author Response · Authors · 2025-11-20
>
> Dear Reviewer ADkH,
>
> Thank you very much for your thoughtful and positive review. We appreciate your recognition of the importance of jointly studying persuasiveness and susceptibility, the clarity of the dual-role simulation, and the value of our human-alignment validation. We respond below to your main concern regarding the role of LLMs as proxies for human persuasion.
>
> ---
>
> ### **(W1) On LLMs as Proxies for Human Discourse**
>
> Thank you for raising this important point. We fully agree that LLM-to-human persuasion involves dynamics that cannot be perfectly captured by LLM-to-LLM interactions. However, **PMIYC is not designed to replace or approximate human persuasion, but instead targets a distinct and increasingly relevant setting: persuasion within multi-agent LLM systems**. As stated in our paper:
>
> > “As multi-agent systems become more prevalent, persuasion will become an essential skill in agent-to-agent interactions, and LLM persuasiveness and susceptibility to persuasion will become increasingly critical.”
>
> LLMs will increasingly act in teams, coordinate across tools, and critique or refine one another’s reasoning. Understanding how these systems influence each other, and how susceptible they are to harmful persuasion (for example, the propagation of misinformation or jailbreak attempts), is essential for safety. Human-only studies do not scale to these multi-turn, cross-model settings.
>
> Regarding real-world persuasion datasets such as ChangeMyView or PersuasionArena: these resources are valuable, but they cannot be directly integrated into PMIYC because our evaluation protocol requires the persuadee to dynamically express its own evolving opinion throughout a multi-turn conversation, conditioned on the persuader’s previous turns. Human-written corpora provide only static, one-sided dialogues, which breaks PMIYC’s turn-by-turn tracking of belief updates and prevents standardized cross-model comparisons.
>
> To mitigate the gap between LLM-only and human persuasion, we additionally conduct two forms of alignment validation. Both the human stance annotations and action-based decision tests show strong agreement with PMIYC’s outputs. These results suggest that, while PMIYC is not intended to model human persuasion, it yields meaningful and behaviorally grounded insights that can inform alignment and multi-agent safety.
>
> ---
>
> Thank you again for your constructive review and for highlighting the potential of PMIYC as a scalable framework for studying persuasion dynamics. If you have any additional questions or points that we can further clarify, please let us know. If our responses satisfactorily address your concerns, we would be grateful if you would consider reflecting this in your scores.

---

### Official Review · Reviewer_WcXZ · 2025-11-02

**Soundness:** 3
**Presentation:** 2
**Contribution:** 3
**Rating:** 4
**Confidence:** 3

**Summary:**

This paper presents PMIYC (Persuade Me If You Can), an automated framework to evaluate both persuasive effectiveness and susceptibility to persuasion in LLMs through simulated multi-turn conversations between two agents, a persuader and a persuadee. The framework measures opinion shifts using a normalized change in agreement (NCA) score, and experiments span subjective debates and misinformation contexts across multiple state-of-the-art models. The experimental results illustrate that models like Llama-3.3-70B and GPT-4o are strong persuaders, but GPT-4o resists misinformation far better, showing over 50% greater robustness. PMIYC also validates its self-reporting approach with human annotations and multiple consistency checks. In conclusion, it is a large-scale, scalable, and empirically grounded attempt to systematically study persuasion dynamics in LLMs.

**Strengths:**

The paper introduces a well-structured, scalable, and reproducible framework that replaces costly human evaluation while maintaining strong alignment with human judgments.

It investigates both sides of persuasion, i.e., how models persuade others and how they can be persuaded, addressing a critical but underexplored angle in AI safety and robustness.

The experimental design is comprehensive, covering single- vs multi-turn setups and subjective vs misinformation contexts, showing good methodological maturity.

Validation through human annotation and multiple consistency checks (self-reports, MCQs, LLM-as-judge) adds credibility to the findings.

**Weaknesses:**

The framework still relies on LLM self-reports as the basis for measuring belief change, which, even though partially validated, remains an imperfect and indirect proxy for true persuasion; models might simulate agreement rather than genuinely “change” stance.

The persuasion domains (subjective and misinformation claims) are limited; adding more diverse or higher-stakes contexts like moral dilemmas, political reasoning, or multi-agent negotiations could strengthen generality.

The turn-based setup is rigid and scripted, lacking elements of conversational flow or adaptive context that occur in human persuasion; this limits ecological realism.

Although human annotations are used for validation, the sample size (125 conversations) and annotator pool (12 graduate students) are relatively small compared to the scale of automated experiments.

The paper highlights ethical risks but does not deeply discuss dual-use implications; for instance, how this framework could unintentionally help design more manipulative persuasion systems if misused.

**Questions:**

How “agreement shifts” in the misinformation setting distinguish between semantic clarification (e.g., revising a claim) and genuine persuasion toward falsehoods.

The normalization formula for NCA is mathematically clean, but it’s unclear how sensitive it is to initial scores near neutrality (e.g., starting at 3).

Whether the persuadee’s “final decision” prompt creates anchoring or framing effects that bias final agreement scores.

How conversation failures (models refusing to answer or deviating from roles) were handled beyond generation success rates—did these get excluded or repaired?

The relationship between persuasion effectiveness and linguistic strategies (e.g., emotional appeals, evidence use) is mentioned but not analyzed; including such qualitative analysis would make the results more interpretable.

**Details Of Ethics Concerns:**

This work studies LLMs’ persuasive and susceptibility behaviors, including in misinformation contexts, which carries dual-use and safety risks. The framework could potentially be misused to develop manipulative AI systems. It also involves human annotators, warranting a quick check for responsible research and data-handling practices.

---

> ### Author Response · Authors · 2025-11-20
>
> Dear Reviewer WcXZ,
>
> Thank you very much for your thoughtful and careful review. We sincerely appreciate your recognition of the strengths of PMIYC, including its scalable multi-agent design, dual-perspective evaluation of persuasion and susceptibility, comprehensive experimental setups, and validated alignment with human judgments. We address your concerns and questions in detail below.
>
> ---
>
> ### **(W1) Reliance on Self-Reports as a Measure of Belief Change**
>
> Thank you for raising this important point. We agree that self-reported agreement is an imperfect proxy for “true” belief change. For this reason, we designed PMIYC to include multiple, independent consistency checks, not only self-reporting. Specifically:
>
> - **Action-based MCQ tests**, which reveal the model’s underlying stance when it must commit to a discrete answer. In the misinformation domain, these MCQs additionally test whether persuasion causes the model to act on the false claim, for example by selecting the incorrect option when it reports agreeing with the misinformation.
> - **LLM-as-judge consistency**, which evaluates whether an external model interprets the dialogue as expressing a shift.
> - **Human annotations**, which show 76% agreement with model self-reports at the turn level.
>
> MCQs represent the ideal verifiable signal, since they reflect what the model does, not just what it states. Importantly, we find that MCQ decisions strongly correlate with self-reports across domains, which indicates that self-reporting can serve as a reliable and scalable proxy for tracking persuasion dynamics in large multi-agent experiments.
>
> We will clarify this framing in the camera-ready version and emphasize that self-reports are treated as one component within a multi-faceted evaluation pipeline rather than the sole basis for measuring belief change.
>
> ---
> ### **(W2) Limited Persuasion Domains**
>
> We appreciate this comment. We grouped the experiments into “subjective” and “misinformation” settings for analytical clarity, but each contains substantial internal diversity.
>
> Through some manual, and LLM-based classification, we see that subjective claims span more than 10 high-level categories, for example:
>
> - Government & Public Policy (320)
> - Ethics, Society & Culture (275)
> - Technology & Digital Governance (155)
> - Education (70)
> - Human Rights & Civil Liberties (40)
> - Law & Justice (30)
> - Health & Medicine (15)
> - Environment & Climate (20)
>
> Misinformation claims are derived from TruthfulQA, which includes 38 categories such as health, law, finance, history, statistics, and political misinformation.
>
> We will add a clearer breakdown in the final version. We also agree that additional domains, such as the ones suggested, would further enrich PMIYC, and we view these as natural extensions of our framework.
>
> ---
>
> ### **(W3) Rigid Turn-Based Interaction Pattern**
>
> We acknowledge that our turn-based dialogue protocol simplifies conversational flow. The choice was intentional: **PMIYC is designed for LLM-to-LLM persuasion**, where agents in multi-agent systems typically follow structured turn boundaries and cooperative protocols rather than exhibit human-like interruption or overlapping speech.
>
> Despite the rigidity, PMIYC reveals rich persuasion dynamics, including turn-by-turn belief drift, end-of-conversation reversals, and susceptibility accumulation. In the final version of our paper, we will emphasize that naturalistic conversational variability—disfluencies, turn overlaps, backchanelling, or uneven turn lengths—is an interesting direction for future work, particularly to study whether such properties increase or decrease susceptibility.
>
> ---
>
> ### **(W4) Human Annotation Sample Size**
>
> Although the annotation set is modest (125 conversations), each conversation spans multiple turns (591 utterances total) and requires nuanced stance identification under persuasive pressure, which is cognitively demanding for human annotators. Together with the MCQ tests and LLM-as-judge evaluations, these additional signals consistently support the reliability of self-reports within PMIYC. Moreover, the agreement between humans and models yields a Cohen’s κ of 0.63 for the three-way judgments, which reflects substantial agreement for a task of this complexity. We will include these additional comments in the final manuscript and clarify that our human annotation study is intended as a validation sample, to verify the fidelity of self-reported stance changes, rather than a full-scale human persuasion experiment.
>
> ---
>
> continued...

---

> > ### Author Response · Authors · 2025-11-20
> >
> > ### **(W5) Dual-Use and Ethical Implications**
> > Thank you for highlighting this critical issue. We agree that a deeper discussion of dual-use concerns will strengthen our work. We will expand the brief discussion in our Ethics Statement on line 484 by adding the following:
> > > While the goal of PMIYC is to evaluate persuasion and susceptibility for safety purposes, it is important to acknowledge that the same insights could, in principle, be misused to design more persuasive or more rigid models. At the same time, responsible deployment of LLMs requires measuring these behaviors. Developers need tools that reveal whether a model is overly persuasive, unusually susceptible, or dangerously resistant to factual correction before the model is released. PMIYC is therefore intended as an evaluation framework that supports red-teaming, alignment diagnostics, and the construction of robustness benchmarks. We encourage future work to apply PMIYC toward improving model resilience to harmful persuasion rather than amplifying manipulative capabilities.”
> > We believe the most responsible path forward is to enable reliable evaluation of persuasive and vulnerable behaviors, especially as LLMs increasingly operate in multi-agent and autonomous settings where persuasive influence can emerge unintentionally.
> > ---
> >
> > ### **Responses to Specific Questions**
> >
> > #### **(Q1) Distinguishing Clarification vs. Genuine Persuasion in Misinformation**
> > Thank you for highlighting this nuance. PMIYC minimizes semantic drift by:
> > (i) re-presenting the *exact original misinformation claim* in the final decision prompt (Sec. 3.1 and Figure 27)
> > (ii) validating self-reports via human and LLM-judge alignment (95% subjective, 87% misinformation; Appendix C and E), and
> > (iii) using action-based MCQs. In misinformation, persuasion is counted as *genuine* only when the persuadee both reports support and selects the incorrect target answer (GPP = 64–78%; Table 2)
> > Together, these steps ensure shifts reflect real persuasion toward falsehoods, not just benign clarification.
> >
> > ---
> >
> > #### **(Q2) NCA Sensitivity Near Neutrality**
> > NCA normalizes by the maximum possible change in either direction, and around neutrality (score = 3) this range is symmetric (±2). This avoids instability and ensures comparability across starting positions (Sec. 3.2) for persuadee susceptibility. Appendix G shows that absolute (non-normalized) shifts lead to the same qualitative conclusions for persuaders; the results only differ for the persuadees across domains.
> >
> > ---
> >
> > #### **(Q3) Anchoring Effects from the Final Decision Prompt**
> > The final decision uses a neutral prompt that restates the original claim (Sec. 3.1, Figure 27). To test anchoring, we compare final self-reports with action-based MCQs that use entirely different wording; these match in >90% of cases for most models (Table 2), suggesting minimal anchoring. Additionally, in misinformation cases, persuadees often correct earlier drift when reminded of the original claim (Sec. 5.1, Figure 6).
> >
> > ---
> >
> > #### **(Q4) Handling Conversation Failures**
> > We retry generation up to five times for refusal or formatting deviations (Appendix B.0.1). Persistent failures lead to exclusion for that specific model–claim pair only; no outputs are edited or repaired. Success rates remain high (96% subjective; 99% misinformation), with only Claude 3 Haiku showing higher refusal rates.
> >
> > ---
> >
> > #### **(Q5) Lack of Linguistic Strategy Analysis**
> > We appreciate this suggestion and agree that this is a valuable direction. While PMIYC focuses on scalable measurement, it already surfaces strategy-related patterns. For example, persuaders are more effective when they personally support the claim (Figure 7) and more persuasive and less susceptible through deeper reasoning (Appendix F, check results with o4-mini). A deeper rhetorical or argument-style analysis is a natural extension, and we will mention this direction in the discussion.
> >
> > ---
> >
> > Thank you again for your thorough and constructive review. Your feedback has helped us refine the framing, clarify methodological choices, and strengthen the discussion of limitations and ethics. If you have any additional questions or points that we can further clarify, please let us know. If our responses satisfactorily address your concerns, we would be grateful if you would consider reflecting this in your scores.

---

### Official Review · Reviewer_YFJz · 2025-11-02

**Soundness:** 2
**Presentation:** 2
**Contribution:** 2
**Rating:** 2
**Confidence:** 4

**Summary:**

This paper introduces PMIYC (Persuade Me If You Can), an automated framework for evaluating both persuasive effectiveness and susceptibility to persuasion in large language models through multi-agent conversational interactions. The framework simulates multi-turn dialogues between persuader and persuadee agents, tracking opinion changes via a normalized change in agreement (NCA) metric across diverse settings, including subjective claims and misinformation scenarios. The authors show several findings through experiments.

**Strengths:**

- The paper proposes an automated framework to evaluate the persuasiveness and susceptibility to persuasion.
- The paper provides comprehensive experimental settings that span multiple dimensions, including single-turn vs. multi-turn, subjective vs. misinformation, different model families and sizes.

**Weaknesses:**

- The paper lacks solid technical contributions, as well as interesting insights and findings. There are already well-established works that measure the persuasiveness of language models [1]. This work seems to be an incremental extension of prior works.
- It’s not realistic to only use LLMs to simulate persuasion conversations, as there could be significant gaps in the behaviors in persuasion between humans and LLMs.
- The paper did not extensively discuss the sycophancy biases in LLMs [2]. The LLMs are prone to following and accepting requests.

[1] Measuring the Persuasiveness of Language Models. Anthropic. 2024.

[2] Towards Understanding Sycophancy in Language Models. Anthropic. 2023.

**Questions:**

What are the significant implications or takeaways from the paper?

---

> ### Author Response · Authors · 2025-11-20
>
> Dear Reviewer YFJz,
>
> Thank you for taking the time to review our work and for recognizing PMIYC’s automated evaluation framework and the breadth of experimental settings we explore. We address your concerns below.
>
> ---
>
> ### **(W1) Technical Novelty**
>
> We appreciate the connection to Anthropic’s “Measuring the Persuasiveness of Language Models” [1]. As we note in our paper, this work indeed inspired part of our motivation to study persuasion systematically. However, PMIYC takes a substantively different direction. Prior work primarily evaluates model-to-human persuasiveness using human annotators, typically in single-turn or non-interactive settings. In contrast, PMIYC introduces a framework built specifically for *multi-agent, multi-turn* persuasion and for evaluating both persuasive effectiveness and susceptibility to influence.
>
> Our framework, PMIYC, differs from [1] in several key ways:
>
> 1. Models persuasion as an interactive, iterative process between two LLMs. This enables analysis of turn-by-turn belief dynamics, early-turn influence, and cumulative persuasion patterns.
>
> 2. Provides a dual-perspective evaluation that measures not only how persuasive a model is, but also how susceptible it is as a persuadee. Susceptibility is central to safety and jailbreak risk, yet it is not addressed in [1].
>
> 3. Incorporates both self-report validation and action-based evaluation, demonstrating 76% human–model alignment on stance annotations (Cohen’s κ = 0.63) and over 90% consistency between self-reported and action-based decisions. This goes beyond the evaluation methodology used in prior work.
>
> Through these differences, PMIYC enables the study of phenomena that one-shot or human-only evaluations cannot capture, such as multi-turn belief drift, domain-specific vulnerability (Figure 5(b)), and end-of-turn reversals. We therefore view our work as complementary to [1] and as addressing a distinct and important gap in evaluating persuasion and susceptibility in multi-agent LLM systems.
>
> ---
>
> ### **(W2) Human vs. LLM Persuasion**
> We agree that LLM-to-human persuasion involves complexities not fully captured by LLM-to-LLM interactions. However, PMIYC is not intended to replace human-agent evaluation. Its purpose is to evaluate *multi-agent* persuasion dynamics, which are increasingly central to emerging LLM systems.
>
> As stated in our paper (lines 50-52):
>
> > As multi-agent systems become more prevalent, persuasion will become an essential skill in agent-to-agent interactions, and LLM persuasiveness and susceptibility to persuasion will become increasingly critical.
>
> PMIYC addresses a gap in evaluating LLM-to-LLM persuasion, where human annotation is expensive and does not scale to multi-turn, cross-model experiments. PMIYC enables systematic cross-model comparisons, large-scale multi-turn analysis, and measurement of susceptibility, which is a crucial factor in multi-agent safety.
>
> ---
>
> ### **(W3) Sycophancy and Persuasion**
>
> Thank you for highlighting this connection. Sycophancy is indeed related to persuasion. We will clarify in the camera-ready version that PMIYC can surface sycophancy-like behavior by identifying patterns such as over-agreement in subjective scenarios, end-of-turn reversals where the persuadee aligns with the persuader’s reasoning (Section 5.1), and domain-dependent over-compliance. In addition, PMIYC provides a natural platform for examining why sycophancy correlates with susceptibility, which aligns with Reviewer aUsk’s suggestion to explore causal factors in post-training.
>
> ---
>
> ### **(Q1) Key Implications and Takeaways**
>
> PMIYC yields several significant, actionable insights:
> - Susceptibility emerges as a measurable alignment risk, especially in misinformation and jailbreak settings.
> - Multi-turn persuasion dynamics can be evaluated at scale, showing that susceptibility increases over turns. This suggests that multi-turn resilience should be incorporated into alignment and safety training.
> - Domain-sensitive behavior is observed, for example, GPT-4o shows strong resistance to misinformation persuasion while remaining flexible on subjective claims.
> - PMIYC’s behavioral outputs can be used directly for reward shaping, preference dataset construction, belief-calibration objectives, and multi-turn adversarial training.
> - Persuader effectiveness depends on the model’s internal stance (Figure 7), which suggests that belief calibration may be valuable as a training objective.
>
> ---
>
> Thank you again for your review. We hope this clarifies the novelty and value of PMIYC and its relevance to both multi-agent systems and LLM alignment. If there are any remaining questions or points we can further clarify, please let us know. If our responses satisfactorily address your concerns, we would be grateful if you would consider reflecting this in your score.

---

### Official Review · Reviewer_aUsk · 2025-11-06

**Soundness:** 3
**Presentation:** 3
**Contribution:** 2
**Rating:** 4
**Confidence:** 3

**Summary:**

This paper introduces Persuade Me If You Can (PMIYC), a framework for evaluating persuasion effectiveness and susceptibility among large language models (LLMs). It simulates multi-turn dialogues between a persuader and persuadee LLM to measure how effectively one can convince the other and how resistant models are to persuasion. The framework introduces a Normalized Change in Agreement (NCA) metric to quantify persuasion outcomes and validates self-reported results through human annotations and action-based tests. Experiments across various models (GPT-4o, Llama-3.3-70B, Claude-3 Haiku) show that multi-turn interactions increase persuasion success, larger models are more persuasive, and susceptibility varies by context, especially under misinformation.

**Strengths:**

1. PMIYC is the first scalable framework to evaluate both persuasion and susceptibility in LLMs using fully automated, multi-turn conversations—an advancement over prior one-shot or human-only methods.

2. The methodology is solid, featuring the novel NCA metric and strong validation through human annotations (>75% alignment) and behavioral consistency (>90%).

3. The paper is well-structured, with clearly explained roles, metrics, and experimental design.

**Weaknesses:**

Limited Causal Analysis of Model Behavior
While the paper provides clear empirical findings (e.g., that Llama-3.3-70B and GPT-4o exhibit similar persuasive effectiveness), it does not explore why models behave this way as persuaders or persuadees. The discussion remains descriptive rather than analytical. Readers are left without insight into what underlying factor, such as architectural differences, pretraining data quality, or post-training methods like RLHF, might explain variations in persuasive ability or resistance. A more diagnostic analysis could connect observed behaviors to known model design or alignment techniques, helping contextualize the findings beyond surface-level comparisons.

The paper successfully measures persuasion and susceptibility but stops short of deriving actionable implications for safer LLM development. It does not clarify how insights from PMIYC could inform strategies to reduce susceptibility to misinformation or balance persuasiveness with ethical safety. For instance, should future LLMs be optimized for higher factual resistance, calibrated confidence, or improved self-consistency? Including a discussion of how PMIYC results can guide alignment, training objectives, or dataset design would significantly enhance the work’s applied impact.

**Questions:**

Can you try to answer my questions in the weakness section?

---

> ### Author Response · Authors · 2025-11-20
>
> Dear Reviewer aUsk,
>
> Thank you very much for your thoughtful and encouraging review. We sincerely appreciate your positive assessment of PMIYC’s contributions, scalability, and validation methodology. Below, we address your two questions in the weakness section.
>
> ---
>
> ### **(W1) Causal Analysis of Model Behavior**
>
> Thank you for raising this excellent point. We fully agree that understanding why different models behave as they do, whether due to architecture, pretraining data, or RLHF, is an important direction. In our current evaluation, the mixture of closed-source and heavily instruction-tuned open-weight models limits our ability to perform such causal attribution.
>
> In the camera-ready version, we will clarify this limitation and explicitly discuss how PMIYC can support controlled causal studies when applied to families of fully open-source models. We are enthusiastic about following up on this as separate future work, for example, by varying RLHF strength, alignment data, or instruction-tuning procedures within the same base model and comparing the resulting behavior under PMIYC. We truly appreciate this suggestion and believe it highlights a valuable next step enabled by our framework.
>
> ---
>
> ### **(W2) Actionable Insights for Alignment and Safety**
>
> Thank you for pointing this out. We agree that PMIYC’s findings can be connected more explicitly to practical implications for building safer and more robust LLMs, and we will add this discussion as such:
>
> > PMIYC provides a structured approach to integrating persuasion dynamics directly into the LLM training pipeline, offering actionable insights for safer model development. Its domain-sensitive evaluation distinguishes models that appropriately resist misinformation persuasion while remaining flexible on subjective claims (Figure 5(b)), illustrating how future systems could be optimized for higher factual resistance without drifting into over-refusal. This separation enables domain-aware reward objectives in which models are reinforced for factual robustness while maintaining ethically appropriate persuasiveness. PMIYC’s ability to reveal how susceptibility accumulates across turns (Figure 6) supports training strategies that penalize multi-step belief drift, allowing multi-turn transcripts to serve as adversarial episodes for RLHF or preference optimization and informing dataset design focused on reducing gradual misinformation vulnerability. The finding that persuasive strength depends on a persuader’s internal agreement (Figure 7) motivates confidence-calibrated and self-consistency-aware generation objectives, where models stabilize their internal stance before producing arguments to avoid amplifying unsupported claims. Additionally, appendix results showing o4-mini’s high persuasion and strong resistance (Figures 17–18) suggest that targeted reasoning-oriented training may enhance ethical persuasion while reducing susceptibility. Across all settings, PMIYC yields interpretable behavioral signatures, such as robustness profiles and susceptibility asymmetries, that can be used to construct targeted preference datasets, define reward-shaping signals, and evaluate whether new tuning procedures steer models toward desired persuasion behaviors. In this way, PMIYC serves not only as an evaluation tool but as a practical foundation for designing alignment objectives, training priorities, and dataset construction aimed at safer, more calibrated, and more self-consistent LLMs.
>
> ---
>
> Thank you again for your constructive feedback and insightful questions. If you have any additional questions or points that we can clarify, please let us know. If our responses address your concerns, we would be grateful if you would consider adjusting your score accordingly.

---

### Author Response · Authors · 2025-12-02
**Revision to Paper Draft**

Dear Reviewers, ACs, PCs, and Conference Organizers,

Thank you for the time and care you invested in evaluating our submission. We have revised the paper (both main text and the appendix) to address the questions and concerns raised during the rebuttal period. In the updated draft, changes marked in blue respond to Reviewer aUsk, changes in green respond to Reviewer YFJz, and changes in red respond to Reviewer WcXZ’s comments (Reviewer ADkH’s comments did not require modifications to the paper).

The primary areas addressed include:

- **LLM-to-LLM Persuasion:** We clarified that our framework is not intended to replicate human–LLM persuasion, but instead focuses on structured LLM-to-LLM interactions that reflect emerging multi-agent systems. The revisions elaborate on the design motivations for this setting, including the use of coordinated turn-based protocols and their implications.
- **Self-Report Reliability:** We had already conducted complementary MCQ-based evaluations, LLM-as-judge assessments, and human annotations; in the revision, we clarify these methodologies, expand the descriptions, and report additional details to more clearly demonstrate the reliability of self-reported agreement.
- **Implications of PMIYC:** We strengthened the connection between our findings and actionable insights for alignment, safety, and training objectives.

We are happy to make any further adjustments if the paper is accepted. Thank you again for your thoughtful feedback and consideration.

---

### Meta-Review · Area_Chair_gKei · 2025-12-24

**Summary:**

This paper presents PMIYC, a new framework for evaluating the persuasion phenomenon between LLMs. It achieves persuasion via simulating multi-turn dialogues between two LLMs, and then quantifies the effectiveness of one convincing the other. Using this framework for evaluation, the authors discover several interesting conclusions.

Overall the reviewers gave negative ratings. The authors have provided detailed responses to their comments. By reading through the paper, reviewers and rebuttal, AC found the rebuttal is not very satisfactory. It leaves several critical concerns unaddressed. Specifically,

1.	Limited causal analysis. The response does not give concrete solutions and conclusions for causal analysis.
2.	Usefulness of insights for ethical safety. Similarly, the response does not give concrete solutions and results about how we can enhance the safety from the insights.
3.	Incremental technical novelty. While we agree that this work is different from existing ones that focus on human-to-LLM persuasion, but there is limited technical contribution to switch the study from human-to-LLM to LLM-to-LLM. Technically it is not challenging.
4.	Evaluations for more scenarios, like sycophancy, has been asked by the reviewers but are still missing.

Given the above points, AC recommended rejection for this paper.

**Reviewer Concerns:**

Some technical clarifications are addressed, but more critical points are still outstanding, like limited causal analysis, insights for ethical safety or alignment, incremental technical novelty, and more evaluations.

**Reviewer Scores:**

Given the above points, AC thinks the reviewers will not adjust their scores.

---

### Decision · Program_Chairs · 2026-01-26

Reject